# Association of Vitamin C Treatment with Clinical Outcomes for COVID-19 Patients: A Systematic Review and Meta-Analysis

**DOI:** 10.3390/healthcare10122456

**Published:** 2022-12-05

**Authors:** Wen Yan Huang, Jiyoun Hong, Sung-Il Ahn, Bok Kyung Han, Young Jun Kim

**Affiliations:** 1Department of Food and Biotechnology, Korea University, Sejong 30019, Republic of Korea; 2BK21 FOUR Research Education Team for Omics-Based Bio-Health in Food Industry, Korea University, Sejong 30019, Republic of Korea; 3Department of Food and Regulatory Science, Korea University, Sejong 30019, Republic of Korea

**Keywords:** vitamin C, COVID-19, mortality, hospitalization, systematic review, meta-analysis

## Abstract

Background: Vitamin C is an essential nutrient that serves as an antioxidant and is known to reduce the inflammatory response associated with pneumonia and acute respiratory distress syndrome in patients with the coronavirus disease (COVID-19), but its clinical effects remain controversial. Methods: This study aimed to investigate the therapeutic effect of vitamin C administration on the clinical outcomes of COVID-19 patients through a systematic review and meta-analysis. Results: Nineteen studies were selected, of which 949 participants administered vitamin C were in the intervention group, and 1816 participants were in the control group. All-cause mortality, hospitalization duration, length of intensive care unit stay, and ventilation incidence in COVID-19 patients were analyzed. The intervention group tends to have a lower risk ratio (RR = 0.81, 95% CI: 0.62 to 1.07; *I*^2^ = 58%; Q = 40.95; *p* < 0.01) in all-cause mortality than the control group. However, there were no significant differences in ventilation incidence, hospitalization duration, and length of ICU stay between the two groups. In the subgroup analysis for all-cause mortality, the risk ratio for RCT as study design, combination therapy, of vitamin C was lower than that of the combination therapy with other agents. A moderate dosage showed a lower RR than a higher dose. Conclusion: The results suggest that vitamin C may lower mortality in COVID-19 patients, but further large-scale studies are required to assess the role of vitamin C in the treatment of COVID-19.

## 1. Introduction

As reported by the World Health Organization, coronavirus disease 2019 (COVID-19) has infected more than 625 million individuals worldwide, with approximately 6.5 million deaths by October 2022 [1]. COVID-19, caused by severe acute respiratory syndrome coronavirus-2 (SARS-CoV-2), was initially considered to spread primarily by droplets and direct contact [2,3]. The clinical outcomes of COVID-19 range from absent or mild symptoms, such as cough, sore throat, headache, and loss of taste or smell, to severe respiratory illness, such as inflammatory cytokine storm syndrome or even death [4,5,6]. Some of the distinguishing features of a more severe COVID-19 infection course include the development of acute respiratory distress syndrome (ARDS), secondary infections such as bacterial pneumonia, and severe sepsis [7,8]. COVID-19 infection-induced pulmonary injury with ARDS and septic shock has increased hospitalization days and mortality in patients aged ≥60 years [9,10]. A previous meta-analysis reported that 32.8% of infected patients had ARDS, 20.3% were transferred to the intensive care unit (ICU), and 13.9% died [11]. Thus, patients with severe symptoms or systemic complications require intensive care to reduce the mortality risk. 

In the early COVID-19 outbreak, quick effort led to the discovery of several promising drug treatments for patients such as remdesivir, paxlovid, and dexamethasone [12]. Under the prolonged COVID-19, however, due to increased public interest in dietary supplements or nutraceuticals, whether and to what extent they can provide any preventive or therapeutic benefit against COVID-19 has emerged as an important issue among scientists. Especially, the onset of the COVID-19 pandemic has seen a surge in nutraceutical sales despite the lack of evidence to support the use of vitamins and other supplements [13]. Among them, vitamin C (i.e., ascorbic acid) is well known as a potent antioxidant, anti-inflammatory, and antiviral agent and is effective in treating the influenza virus infection caused by common colds [14,15,16].

Generally, vitamin C is administered orally or intravenously. Oral administration of vitamin C is poorly absorbed by the body, making it difficult to achieve therapeutic plasma levels [17]. In contrast, intravenously administered vitamin C can reach the therapeutic level quickly, achieving 30–70 times higher peak plasma concentration by bypassing the limits of intestinal transporters [18]. High-dose intravenous vitamin C (HDIVC) trials based on previous positive clinical outcomes have been used to treat COVID-19 infection [19,20,21]. A randomized clinical trial comparing the effects of HDIVC with control (placebo) in severe COVID-19 patients in 2021 reported the efficacy of HDIVC in mitigating the arterial partial pressure of O_2_/fraction in the inspired O_2_ ratio, whereas there was less correlation with hospital mortality [22]. However, a pilot randomized trial conducted by Darban et al. reported no significant difference in the length of ICU stay between the standard and control groups [23]. In a meta-analysis of randomized controlled trials by Rawat et al. [24], vitamin C treatment did not reduce mortality, hospitalization duration, or length of ICU stay. Another meta-analysis by Huang et al. [25] reported dissimilar results in that HDIVC in pneumonia patients reduced the length of ICU stay and suggested that HDIVC could be a suitable treatment for COVID-19 patients. 

Large-scale clinical trials such as REMAP-AP (NCT02735707), ROVIT(NCT03680274), EVICT-CORONA-ALO (NCT 04344184), and COVID A to Z (NCT 04342728) have been processed to investigate the therapeutic effect of vitamin C against COVID-19, and these results were still mixed. REMAP-CAP trials showed reduced organ support-free days, favoring escalated-dose prophylactic anticoagulation, whereas the ROVIT trial reported adults with sepsis receiving vasopressor therapy in the ICU, and those who received intravenous vitamin C had a higher risk of death or persistent organ dysfunction at 28 days than those who received a placebo [26,27,28,29]. As previous studies have shown similar or contradictory results, the therapeutic effects of Vitamin C, especially HDIVC treatment in COVID-19 patients remain unclear and controversial. Thus, this study aimed to evaluate the possible effects of vitamin C on various clinical outcomes, including all-cause mortality, hospitalization duration, length of ICU stay, and ventilation incidence in patients diagnosed with COVID-19, through a systematic review and meta-analysis using recent studies.

## 2. Materials and Methods

### 2.1. Search Strategy

This review was performed following the Preferred Reporting Items for Systematic reviews and Meta-analyses (PRISMA) guidelines [30]. Electronic searches were conducted in the PubMed, EMBASE, and Cochrane Library databases between January 2019 and June 2022. The pre-defined keywords were used together using “OR” and “AND”, including vitamin C OR ascorbic acid OR ascorbate AND COVID-19 OR coronavirus disease 2019 OR SARS-CoV-2 OR severe acute respiratory syndrome coronavirus 2.

### 2.2. Study Selection

In this study, the PICO (population, intervention, comparison, and outcomes) model for establishing the search strategy was set as follows: (a) patients or population: patients with COVID-19, regardless of sex; (b) intervention: vitamin C, comparison of its administration by mono or combination with other nutrients for subgroup analysis; (c) comparison: placebo or usual care, and (d) outcome: all-cause hospital mortality, hospitalization duration, length of ICU stay, and ventilation incidence were included. Only studies written in English were included in this meta-analysis. Studies were excluded if they were pre-print data or if they did not have the above-described clinical outcomes. When two or more studies referred to the same study, only one article was included in the review. Two authors (W.Y.H and J.Y.H) assessed all identified titles/abstracts for possible inclusion and reviewed the full text against the inclusion criteria. Any disagreement regarding the study was evaluated by a third author (S.I.A).

### 2.3. Data Extraction and Coding

Duplicate studies were excluded and studies that met the eligibility criteria were selected. Data for the meta-analysis were independently coded by two authors (W.Y.H and J.Y.H) and tabulated using an Excel spreadsheet (Microsoft Excel 365, Microsoft Corp., Redmond, Washington, DC, USA). To convert data described in median (range) or median (interquartile range, IQR) to a mean (SD), methods by Luo et al. for median (IQR) and Hozo et al. for the mean (SD) were applied using automatically calculated formulas from a website (https://www.math.hkbu.edu.hk/~tongt/papers/median2mean.html (accessed on 15 June 2022) [31,32,33].

### 2.4. Risk of Bias Assessment and Publication Bias

The risk of bias was evaluated with Cochrane’s risk of bias tool for RCTs (RoB 2.0) tool for randomized controlled trials (RCTs) on five domains (randomization process, deviations from intended interventions, missing outcome data, measurement of the outcome, and selection of the reported result) as ‘low risk’, ‘some concern’, and ‘high risk’ [34]. The Newcastle-Ottawa quality assessment scale was used for non-RCT studies, with study quality ranging from low (0–3), moderate (4–6), and high (7–9), respectively [35]. The assessment was performed independently by two reviewers (W.Y.H and J.Y.H), and disagreements during the evaluation were solved by a discussion with the other researcher (S.I.A) until approaching a common opinion. Funnel plots and Egger’s linear regression test were used to evaluate publication bias. The trim-and-fill method was used to compensate for potential publication bias. The results adjusted by the trim-and-fill method can be used as a sensitivity analysis to help identify the possible impact of publication bias in a meta-analysis [36,37].

### 2.5. Statistical Analysis

To calculate the effect size of the studies included in the meta-analysis, all-cause mortality and ventilation incidence were analyzed by extracting the risk ratio (RR), and hospitalization duration and length of ICU stay were analyzed by extracting the standardized mean difference (SMD). A forest plot with 95% effect estimates or confidence intervals (CI) was also plotted to test for publication bias in the meta-analysis. Random-effects were applied to calculate the pooled effect estimates considering the underlying variations across the included trials. Heterogeneity was assessed using the *I*^2^ index and Q statistics. Significant heterogeneity was defined as *I*^2^ > 50.0% or *p* < 0.10 [38]. The categorical and continuous data were analyzed by meta-ANOVA and meta-regression, respectively. Funnel plot and Egger’s test was used to confirm the publication bias of the studies included in the meta-analysis. In the result of Egger’s test, in cases where the *p*-value was >0.05, it was considered that there was no publication bias. All data analyses were performed using R statistical program version 4.1.3 (The R Foundation for Statistical Computing, Seoul, Republic of Korea), and *p* < 0.05 was considered statistically significant.

## 3. Results

### 3.1. Study Selection

Figure 1 illustrates the procedure of the meta-analysis. As a result of searching articles, 1007 studies were initially identified from the various online databases. After reviewing the titles and abstracts and assessing their eligibility through the whole text based on inclusion and exclusion criteria, 94 studies were excluded due to the case that the documents had no outcomes (n = 48), irrelevant outcomes (n = 34), or were not peer-reviewed documents (n = 12). Finally, 19 studies were included in the meta-analysis. 

### 3.2. Study Characteristics of Included Studies

Details of the study characteristics are summarized in Table 1. Ten studies (52.6%) were randomized controlled trials, and nine (47.4%) were non-RTCs as retrospective studies. Nine hundred and forty-nine participants administered vitamin C were included in the intervention group, and 1816 participants were in the control group. The sample sizes of each group ranged from eight to 153 and 10 to 558, respectively. The dosage of vitamin C per day went from 500 mg to 24 g; the dosage in Krishnan et al. [39] was not described in the text. Vitamin C was intravenously administered in 15 studies and orally administered in three. The duration of vitamin C treatment ranged from four days to 14 days, but six studies were not described precisely. Seven studies used monotherapy of vitamin C, and eleven used combination therapy with various agents including other vitamins, zinc, thiamin, and steroids. Except for Hakamifard’s study [40], 18 studies had all-cause mortality outcomes, then were subdivided into two groups to investigate the difference in study design (RCT vs. non-RCT), treatment methods (monotherapy vs. combination therapy), route (IV vs. Oral), and dosage (moderate vs. high).

### 3.3. Risk of Bias Assessment in Individual Studies

The result of the risk of bias assessment for the included RCT studies was indicated in Figure 2. All studies showed a low risk of bias in domain 3 (bias due to missing outcome data). Two studies [23,51] were judged to have a ‘high risk of bias’’, six studies [21,22,42,43,47,50] showed a ‘low risk of bias,’ and two studies [20,40] were found to have an ‘unclear’ overall risk of bias assessment. The high risk of bias is primarily due to domain 2 (bias due to deviations from the intended intervention). The study of Zhang et al. [22] was judged to have a low risk of bias in all domains. The risk of bias in the nine non-RCT students was evaluated using the Newcastle-Ottawa scale (Table 2). Except for Krishnan et al. [39], and Hess [45], the others were judged to be high quality (7–8 stars). The study of Krishnan et al. [39] was evaluated as moderate quality due to the issues from inappropriate case definition, selection of controls, and ascertainment of exposure.

### 3.4. Result of Meta-Analysis

Figure 3 shows a forest plot of the main outcomes. All-cause mortality was assessed in 18 studies involving 2693 participants (911 in intervention and 1782 in control). All-cause mortality amounted to 24.15% vs. 33.95% among participants treated with and without vitamin C, respectively. In the result of the meta-analysis of mortality, a negative effect size was observed in the vitamin C-administered group, but the difference was not significant (RR = 0.81, 95% CI: 0.62 to 1.07; *I*^2^ = 58%; Q = 40.95; *p* < 0.01) (Figure 3a). The RR of the ventilation incidence showed slightly lower than that of the control group, but the difference was not significant (RR = 0.98, 95% CI: 0.77 to 1.25; *I*^2^=0%; Q = 0.56; *p* = 0.97) (Figure 3b). In contrast, the hospitalization duration and length of ICU stay in the vitamin C intervention group was significantly increased compared to that in the control group, but not significant (Figure 3c,d).

Subgroup analyses of all-cause mortality were performed to investigate differences in affected mortality according to the study design, treatment method, route, and vitamin C dosage (Figure 4). In the sub-group result of the study design, RCT showed a lower RR (RR = 0.80; 95%CI: 0.70 to 0.91; *p* = 0.77) than that of non-RCT (RR = 0.85; 95%CI: 0.49 to 1.49; *p* < 0.01; Figure 4a). The RR of monotherapy of vitamin C (RR = 1.07, 95% CI: 0.55 to 2.09) was higher than that of its combination therapy with other agents (RR = 0.72, 95% CI: 0.53 to 0.97), and combination therapy was statistically significant (Figure 4b). Thus, vitamin C administration with other agents in COVID-19 patients could be more effective. The effect sizes of the IV treatment route showed a similar result to the pooled estimates (RR = 0.81, 95% CI: 0.62 to 1.07). Still, there needed to be more evidence comparing the efficacy of the treatment mode, as too few oral studies were included (Figure 4c). The vitamin C dosage was classified as moderate or high based on previous studies [53]. Moderate dosage (>3 g/day) showed a lower RR (RR = 0.76, 95% CI: 0.64 to 0.89) than a high dose (RR = 0.94, 95% CI: 0.48 to 1.86), and the RR in high dose was no significant difference (Figure 4d). Meta-regression was carried out to investigate the effect of vitamin C dose on the hospitalization duration. However, no significant results were obtained (data not shown).

### 3.5. Publication Bias

Publication bias was evaluated by Egger’s linear regression test (Table 3) and funnel plots (Figure 5). Publication biases were not found in all factors in Egger’s test. Because *p*-values ranged from 0.74 to 0.87 for each outcome (*p* > 0.05), the null hypothesis could be rejected. Therefore, significant publication bias was unlikely to occur in Egger’s test. However, publication biases in funnel plots existed in all observed items except hospitalization duration (Figure 5). As shown in Figure 5, the asymmetric distributions were found in all items by the black dots, and the white points corrected the asymmetric distributions. This result does not match the results from Egger’s test. This is thought to be due to the small number of data (less than 10 studies) in most observed items. Table 4 shows the effect size correction results (trimmed effect size) using the trim-and-fill method. For the all-cause mortality, the effect estimate (RR = 0.86, 95% CI: 0.65–1.14) adjusted by adding the assumed missing effect sizes (3 studies) was no different in the direction, size, or statistical significance as compared to the effect estimate before adjustment (RR = 0.81, 95% CI: 0.62–1.07). Hospitalization duration and length of ICU stay have not been added to any studies for adjusting.

## 4. Discussions

The present systematic review and meta-analysis aimed to review the previous evidence for vitamin C administration and determine whether it influenced clinical outcomes compared to the control group in patients with COVID-19. The overall findings indicated that the patients with COVID-19 who were treated with vitamin C manifested signs of reduced RR of all-cause mortality by approximately 19%. In subgroup analyses for mortality, RCT, combination therapy of vitamin C, and moderate dosage showed significant differences in mortality. Ventilation incidence, hospitalization duration, and length of ICU stay by vitamin C treatment did not affect COVID-19 patients. 

Cytokine storm and oxidative stress accelerate ARDS progress in COVID-19, then respiratory failure due to ARDS has been reported as the primary cause of mortality in patients with COVID-19 [54,55]. Vitamin C has been widely used in treating several inflammatory diseases, especially ARDS and sepsis in previous studies [56,57,58]. Numerous attempts have been made to clarify the preventive or therapeutic effects of vitamin C in treating COVID-19. A recent systematic review of RCTs among critically ill patients with sepsis found that IV-VC therapy might be associated with a trend toward reduction in overall mortality [59]. Alternatively, Wang et al. [53] reported that there was no impact on mortality among patients administered low (<3 g/day) or high (≥10 g/day) dosages. However, a moderate dose (3–10 g/day) resulted in a significant mortality reduction of 8.5%. There was no difference in the duration of vasopressor support or ICU/hospital stays between the groups. Lin et al. [60] performed a random-effects meta-analysis of six trials that focused on mortality among patients with sepsis (n = 109). There was no difference in mortality in the overall population, but 40 patients with severe sepsis administered high-dose vitamin C showed a significant reduction in mortality (OR 0.39, 95% CI: 0.16 to 0.94, *p* < 0.05).

In a previous meta-analysis, intravenous administration of vitamin C (IV) in critically ill patients significantly decreased the duration of mechanical ventilation, but no difference was observed in mortality [61]. In the results of a meta-analysis by Langlois et al. [62] vitamin C administration route (oral or IV), therapy method (combination or monotherapy), and dosage (high or low) were similar to our subgroup results. There was no effect on the mortality rates, ICU/hospital stay, or ventilation duration. The meta-analysis by Putzu et al. [63] used only RCTs, and unlike our results, found that vitamin C treatment reduced ICU/hospital stay without affecting mortality. Patients with moderate to severe COVID-19 had significantly increased levels of pro-inflammatory cytokines, systemic inflammation, and multiple organ failure [64]. It has been reported that HDIVC can reduce inflammation in COVID-19 patients and prevent progression to severe conditions or death [65]. In a recent cohort study, HDIVC demonstrated the potential benefit of reducing inflammatory marker levels in patients with severe COVID-19 [66].

Despite some positive therapeutic results in COVID-19 patients, several studies have shown different or contrasting results. The number of completed clinical trials was also insufficient to prove the effectiveness of vitamin C in COVID-19 in the above research results. Therefore, additional randomized controlled studies on vitamin C treatment and follow-up studies on the relevance of various symptoms, target groups, and supply methods are needed. In addition, COVID-19 is a multisystem disease in which oxidative stress is partly responsible for excessive inflammation and circulatory disorders. Vitamin C deficiency has been demonstrated in COVID-19 and other acute severe infections and should also be investigated in long COVID. Further studies about vitamin C treatment on long COVID-associated disease should be investigated in clinical trials. The COVID-19 pandemic has gone through a series of waves for more than two years worldwide as emerging Omicron variants were reported in November 2021 [67]. Because of the development of effective vaccines and oral treatments against the COVID-19 virus and increasing natural or vaccine-induced immunity among populations, many people are expected to return to their daily lives. Patients with critical COVID-19 often require costly mechanical ventilation and membrane oxygenation, which may substantially increase the associated medical costs. As shown in the results of this study, proper use of vitamin C in COVID-19 patients is thought to help reduce mortality and medical expenses.

## 5. Limitation

This study had some limitations. First, a meta-analysis of some included findings, such as IL-6 and CRP levels, could not be performed due to the small number of included studies. Second, subgroup analysis was performed only for all-cause mortality because other outcomes were small sample sizes and could not be divided into sub-group. Third, since all included studies had been previously published, the risk of the file drawer problem may have resulted in publication bias. Lastly, it is important to use caution in applying the findings of this study to all COVID-19 patients because various conditions, such as severe, mild, and asymptomatic, were not considered.

## 6. Conclusions

In this study, vitamin C decreased the risk ratio of all-cause mortality by approximately 19% in COVID-19 patients, in subgroup analyses based on study design (RCT vs. non-RCT), treatment method (monotherapy vs. combination), route (IV vs. oral), and dosage (moderate vs. high). Overall, RCT studies, combination therapy, and administering medium doses effectively improved the condition of COVID-19 patients compared to the control group.

## Figures and Tables

**Figure 1 healthcare-10-02456-f001:**
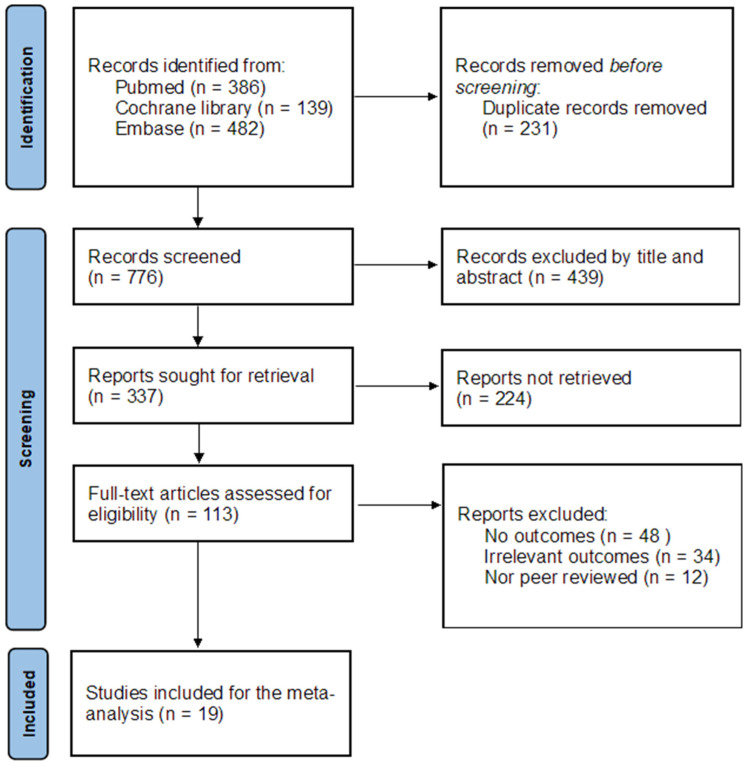
The PRISMA 2020 flow diagram for the meta-analysis.

**Figure 2 healthcare-10-02456-f002:**
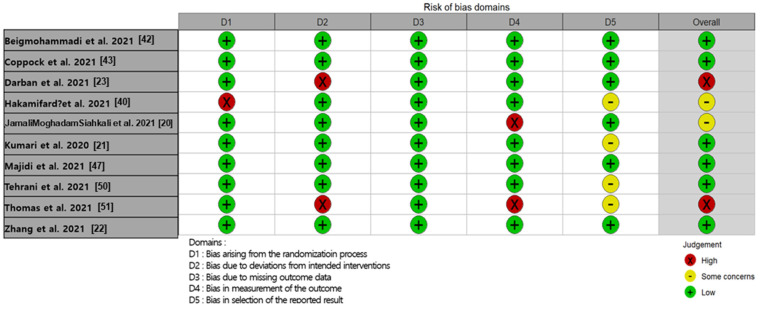
Traffic light plot of risk of bias assessment of included studies using RoB 2.0 criteria, and overall risk of bias.

**Figure 3 healthcare-10-02456-f003:**
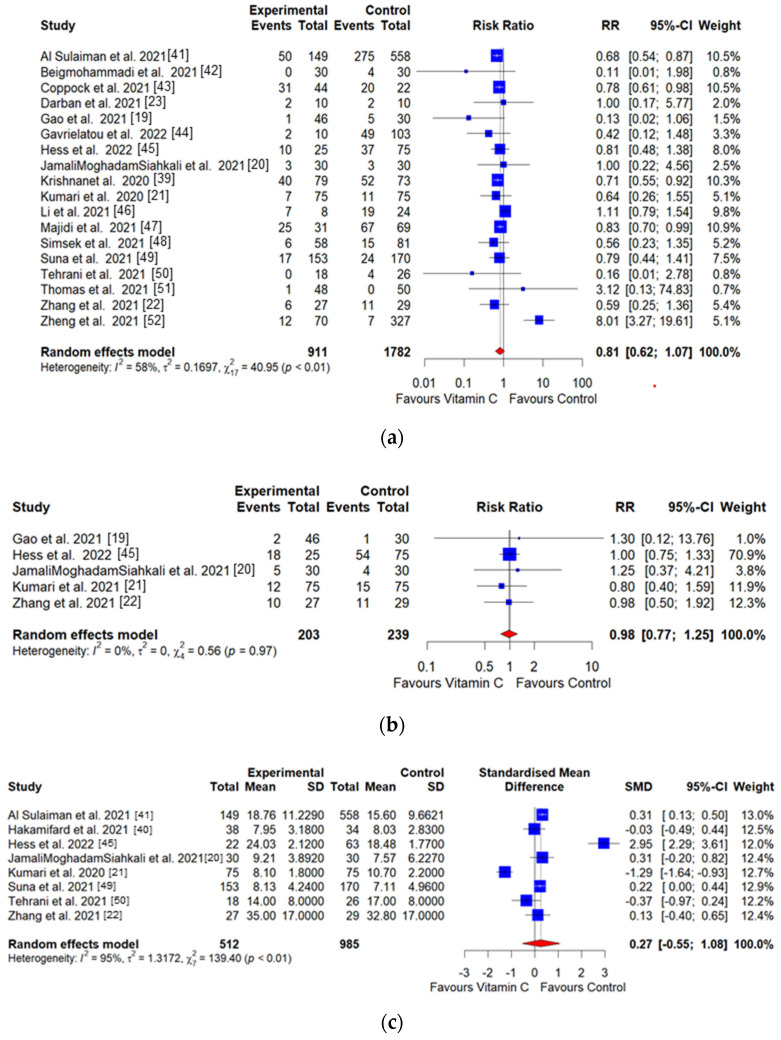
Results of the meta-analysis on the difference in the risk ratio (RR) for (**a**) all-cause mortality, (**b**) ventilation incidence, (**c**) mean difference for hospitalization duration, and (**d**) length of ICU stay between intervention and control groups.

**Figure 4 healthcare-10-02456-f004:**
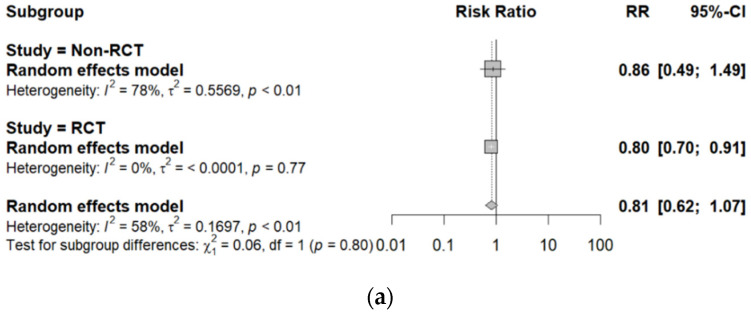
Subgroup analysis for all-cause mortality according to (**a**) study design (non-RCT vs. RCT), (**b**) treatment method (monotherapy vs. combination therapy), (**c**) route (IV vs. oral), and (**d**) dosage (moderate vs. high).

**Figure 5 healthcare-10-02456-f005:**
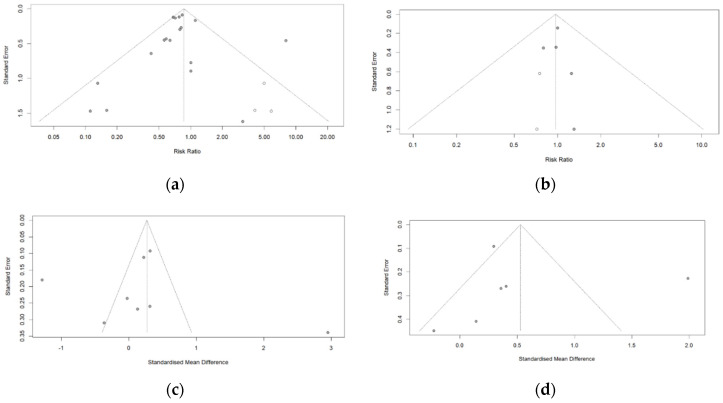
Funnel plots for publication bias evaluation. (**a**) All-cause mortality, (**b**) ventilation incidence, (**c**) hospitalization duration, and (**d**) length of ICU stay between intervention and control groups.

**Table 1 healthcare-10-02456-t001:** Study characteristics of all included in meta-analysis.

No	First Author (Year)	Study Design	Sample Size I/C	Dosage of Vitamin C	Treatment Duration	Treatment Method (Route)	Outcomes Measurement
1	Al Sulaiman (2021) [41]	Retro	149/558	1 g/day	NR	Mono therapy (IV)	All-cause mortality, Hospitalization duration, ICU stay
2	Beigmohammadi (2021) [42]	RCT	30/30	2 g/day	7 days	Combination with vitamin A, D, B1, B2, nicotinamide, pyridoxine, sodium pantothenate (IV)	All-cause mortality
3	Coppock (2021) [43]	RCT	44/22	0.3–0.9 g/kg	5 days	Mono therapy (IV)	All-cause mortality
4	Darban (2021) [23]	RCT	10/ 10	8 g/day	10 days	Combination with melatonin and zinc (IV)	All-cause mortality, ICU stay
5	Gao (2021) [19]	Retro	46/30	12 g/day (1st day), 6 g/day (2nd–5th day)	5 days	Combination with antibiotics, corticosteroids, and other antivirals (IV)	All-cause mortality, Ventilation incidence
6	Gavrielatou (2022) [44]	Retro	10/103	–1.5 g/day	7 days	Combination with thiamine (IV)	All-cause mortality
7	Hakamifard (2021) [40]	RCT	38/34	1 g/day	NR	Combination with vitamin E (Oral)	Hospitalization day
8	Hess (2022) [45]	Retro	25/75	18 g/day	7 days	Mono therapy (IV)	All-cause mortality, ICU stay, Ventilation incidence
9	JamaliMoghadamSiahkali (2021) [20]	RCT	30/30	6 g/day	5 days	Combination with lopinavir and ritonavir (IV)	All-cause mortality, Hospitalization duration, ICU stay, Ventilation incidence
10	Krishnan (2020) [39]	Retro	79/73	NR	NR	Combination with steroids (NR)	All-cause mortality
11	Kumari (2020) [21]	RCT	75/75	50 mg/kg/day	NR	Combination with dexamethasone and prophylactic antibiotics (IV)	All-cause mortality, Hospitalization duration, Ventilation incidence
12	Li (2021) [46]	Retro	8/24	9 g/day	4 days	Combination with hydrocortisone and thiamine (IV)	All-cause mortality, ICU stay
13	Majidi (2021) [47]	RCT	31/69	500 mg/day	14 days	Mono therapy (Oral)	All-cause mortality
14	Simsek (2021) [48]	Retro	58/81	25 g/day	7 days	Combination with hydroxychloroquine, azithromycin, favipiravir (Oral)	All-cause mortality
15	Suna (2021) [49]	Retro	153/170	2 g/day	NR	NR (IV)	All-cause mortality, Hospitalization duration
16	Tehrani (2021) [50]	RCT	18/26	8 g/day	5 days	Mono therapy (IV)	All-cause mortality, Hospitalization duration
17	Thomas (2021) [51]	RCT	48/50	8 g/day	10 days	Mono therapy (IV)	All-cause mortality
18	Zhang et al. (2021) [22]	RCT	27/29	24 g/day	7 days	Combination with antiviral(glucocorticoid) (IV)	All-cause mortality, Hospitalization duration, ICU stay, Ventilation incidence
19	Zheng (2021) [52]	Retro	70/327	4 g/ day	NR	Mono therapy (IV)	All-cause mortality

I, intervention group; C, control group; Retro, retrospective study; RCT, randomized controlled trial; IV, intravenous treatment; ICU, intensive care unit; NR, not reported.

**Table 2 healthcare-10-02456-t002:** Quality assessment by the Newcastle-Ottawa quality assessment scale.

Quality Criteria	Selection				Comparability		Exposure		Total
	Is Case Definition Adequate?	Representativeness of the Cases	Selection of controls	Definition of controls	Comparability on Basis of Design or Analysis	Ascertainment of Exposure	Same Method of Ascertainment for Cases and Controls	Nonresponse Rate	
Al Sulaiman (2021) [41]	★	★	★	★	★	★		★	7
Gao (2021) [19]	★	★	★	★	★	★	★	★	8
Gavrielato (2022) [44]	★	★	★	★	★	★		★	7
Hess (2022) [45]	★	★		★	★		★		5
Krishnan (2020) [39]		★		★	★		★	★	5
Li (2021) [46]	★	★	★	★	★	★	★	★	8
Simsek (2021) [48]	★	★	★	★	★		★	★	7
Suna (2021) [49]	★	★	★	★	★	★	★	★	8
Zheng (2021) [52]	★	★	★	★	★	★	★	★	8

★ means one point.

**Table 3 healthcare-10-02456-t003:** Egger’s linear regression test for publication bias.

Outcomes	Bias	Se. ^1^ Bias	Intercept	Se. Intercept	*t*	df ^2^	*p*-Value
All-cause mortality	−0.10	0.56	−0.21	0.12	−0.17	16	0.87
Ventilation incidence	0.09	0.34	−0.04	0.09	0.28	3	0.80
Hospitalization duration	0.65	3.80	0.05	0.63	0.17	6	0.87
Length of ICU stay	0.94	2.70	0.33	0.50	0.35	4	0.74

^1^ Se: standard error, ^2^ df: degrees of freedom of Q statistic.

**Table 4 healthcare-10-02456-t004:** Trimmed effect size of vitamin C administration in COVID-19 patients.

Items	Added Studies	Adjusted Effect Size	95% CI	*I*^2^ (%)	*p*-Value
All-cause mortality	3	0.86	0.65–1.14	57.4	0.29
Ventilation incidence	2	0.97	0.77–1.22	0.0	0.99
Hospitalization duration	0	0.27	−0.55–1.08	95.0	0.52
Length of ICU stay	0	0.53	−0.10–1.15	90.3	0.10

## Data Availability

Not applicable.

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
