# Peer review of "Association of Vitamin C Treatment with Clinical Outcomes for COVID-19 Patients: A Systematic Review and Meta-Analysis"

_healthcare, 2022, doi:10.3390/healthcare10122456_

Round 1

Reviewer 1 Report

An interesting article covers an important research topic. 

Would benefit from mentioning some of the large RCTs that we are awaiting results from, such as REMAP CAP and LOVITT COVID. The introduction and discussion section does not mention some of the large Sepsis RCTs and more recent SLR/MA on the topic. These sections must be updated in line with the current (albeit rapidly evolving) literature base.

As the literature search was last completed in December 2021 the authors may wish to consider updating the review.

Both the introduction and discussion include some fairly in-depth detail on previous reviews and studies. Whilst this is useful, I think some more generalised statements on these topics may better suit the rapidly evolving evidence base.

Could consider shortening the section on the background of the pandemic.

Lines 82-82 - consider correcting to limited oral bioavailability due to saturation of receptors.

Line 90-91 - consider changing from there was less correlation to the study was underpowered to detect a mortality benefit.

Would benefit from defining the cut-off for high-dose vitamin C. (236)

Consider shortening some of the discussion and removing country-specific mortality rates and guidelines (336). This could be replaced with a shorter, more generalised statement.

361-362 - Need to reference and explain the statement regarding novel variants. I suspect this a wording issue and may be better worded as "to reduce mortality and morbidity from future variants".

The authors may wish to comment on the potential benefits of vitamin C in long COVID.

As the study includes non-RCT studies, the authors should comment on using non-RCT studies in the text. If this is to be published in the current form, then there must be an acknowledgement of the potential bias and a stronger emphasis on the need to interpret the results with caution. Furthermore, there needs to be at least a comment on the weighting of studies according to their risk of bias and study methodology.

I will be happy to provide a timely review of the article following these minor changes and would encourage the authors to re-submit this as soon as possible.

Author Response

Point 1: Would benefit from mentioning some of the large RCTs that we are awaiting results from, such as REMAP CAP and LOVITT COVID. The introduction and discussion section does not mention some of the large Sepsis RCTs and more recent SLR/MA on the topic. These sections must be updated in line with the current (albeit rapidly evolving) literature base.

Response 1: Thanks for the reviewer’s thoughtful comment. We revised the aforementioned comments on lines 74-81.

Point 2: As the literature search was last completed in December 2021 the authors may wish to consider updating the review.

Response 2: We updated the latest review to June 2022 and revised them on lines 93 and Figure 1. For a more detailed search, we used the Embase database instead of Google as shown on Figure 1.

Point 3: Both the introduction and discussion include some fairly in-depth detail on previous reviews and studies. Whilst this is useful, I think some more generalised statements on these topics may better suit the rapidly evolving evidence base.

Response 3: As the reviewer pointed out, we have shortened both the introduction and discussion parts using “Track Change” function.

Point 4: Could consider shortening the section on the background of the pandemic.

Response 4: As the reviewer pointed out, we have shortened the section on the background of the pandemic in the introduction part using “Track Change” function.

Point 5: Lines 82-82 - consider correcting to limited oral bioavailability due to saturation of receptors.

Response 5: Thanks for the reviewer’s thoughtful comment. We have reported a related study (https://doi.org/10.1089/jmf.2020.4743).

Point 6: Consider changing from there was less correlation to the study was underpowered to detect a mortality benefit.

Response 6: As mentioned in point 2, we added total six articles to the meta-analysis through research and review. Also, we made overall modifications and described that he intervention group tends to have a lower risk ratio (RR=0.81, 95% CI: 0.62 to 1.07; I2=58%; Q=40.95; p <0.01) in all-cause mortality than the control group.

Point 7: Would benefit from defining the cut-off for high-dose vitamin C.

Response 7: The criteria of high-dose vitamin C were not exactly determined, but many previous studies reported 1g to 3g per day. Among them, we re-classified dosage referred to a related study (Wang Y, Lin H, Lin BW, Lin JD. Effects of different ascorbic acid doses on the mortality of critically ill patients: a meta-analysis. Ann Intensive Care 2019;9;58.).

Point 8: Consider shortening some of the discussion and removing country-specific mortality rates and guidelines (336). This could be replaced with a shorter, more generalised statement.

Response 8: As the reviewer pointed out, we have shortened the discussion part and removed country-specific information using “Track Change” function.

Point 9: 361-362 - Need to reference and explain the statement regarding novel variants. I suspect this a wording issue and may be better worded as "to reduce mortality and morbidity from future variants"

Response 9: Based on the reviewer pointed out, we've removed the phrase "new variants," which is less relevant.

Point 10: The authors may wish to comment on the potential benefits of vitamin C in long COVID.

Response 10: Thanks for the reviewer’s thoughtful comment. We added the comments on line 320-324.

Point 11: As the study includes non-RCT studies, the authors should comment on using non-RCT studies in the text. If this is to be published in the current form, then there must be an acknowledgment of the potential bias and a stronger emphasis on the need to interpret the results with caution. Furthermore, there needs to be at least a comment on the weighting of studies according to their risk of bias and study methodology

Response 11: We used two tools to evaluate the risk of bias; Cochrane’s risk of bias tool for RCTs (RoB 2.0) for randomized controlled trials (RCTs) and the Newcastle-Ottawa quality assessment scale for non-RCT studies. We conducted a subgroup meta-analysis with RCTs and NON-RCTs (not shown in the article), and there was no difference between the groups. Therefore we might not consider any weighting.

Point 12: I will be happy to provide a timely review of the article following these minor changes and would encourage the authors to re-submit this as soon as possible.

Response 12: Thanks for the reviewer’s thoughtful comment. We look forward to hearing from you regarding our manuscript and thank you again for your time and consideration.

Reviewer 2 Report

The authors wrote this review to determine the prognostic impact of vitamin C administration in COVID-19 patients.

The manuscript was well prepared. Moreover, the cited and their discussion are presented in good style. However, authors have a point that needs to be addressed. 

In this manuscript, we focus on 13 studies. Most of the results discussed in this manuscript seem to be strongly influenced by the results of  Al Sulaiman et al. Is it possible to present supplemental data without Al Sulaiman et al.? Also, can you discuss the weighting of the 13 studies in detail?

Also, as a minor correction, the resolution of all figures and tables is low quality and the text is difficult to read, so please replace all of them with high quality ones.

Author Response

Point 1: In this manuscript, we focus on 13 studies. Most of the results discussed in this manuscript seem to be strongly influenced by the results of Al Sulaiman et al. Is it possible to present supplemental data without Al Sulaiman et al.? Also, can you discuss the weighting of the 13 studies in detail?

Response 1: Thanks for the reviewer’s thoughtful comment. As the reviewer pointed out, we submitted revised supplemental data without Al Sulaiman et al.

Also, we updated the latest review to June 2022 using Embase database instead of Google for a more detailed search and revised them on lines 118 and Figure 1. We made modifications overall results, and we would appreciate a detailed review of the revised ones. 

Point 2: Also, as a minor correction, the resolution of all figures and tables is low quality and the text is difficult to read, so please replace all of them with high quality ones.

Response 2: As the reviewer pointed out, we changed and replaced all low-quality resolution figures and tables.

Supplementary : Review 2

Point 1 : Most of the results discussed in this manuscript seem to be strongly influenced by the results of Al Sulaiman et al. Is it possible to present supplemental data without Al Sulaiman et al.? Also, can you discuss the weighting of the 13 studies in detail?

Reply : As shown in Figure S(a) below, the effect size without AL was not significantly different from the result with AL.

Figure S(b) showed the results with all studies (18 items) in order of weighting. There were43 cases with weights exceeding 10%, and their influence was not considerable.

                (a)

                (b) 

Figure S. Result of the meta-analysis on the risk ratio (RR) for all-cause mortality without Al Sulaiman et al.

Reviewer 3 Report

This paper answers an interesting and relevant question and overall is an important contribution to the literature. The paper is  well executed, though I have several concerns to highlight. 

Data from the literature should be better consulted. For example, what does the current meta-analysis bring new compared to that of 2022: Gavrielatou E, Xourgia E, Xixi NA, Mantelou AG, Ischaki E, Kanavou A, Zervakis D, Routsi C, Kotanidou A, Siempos II. Effect of Vitamin C on Clinical Outcomes of Critically Ill Patients With COVID-19: An Observational Study and Subsequent Meta-Analysis. Front Med (Lausanne). 2022 Feb 11;9:814587. doi: 10.3389/fmed.2022.814587. PMID: 35223911; PMCID: PMC8873176.

Also please see information from: Milani GP, Macchi M, Guz-Mark A. Vitamin C in the Treatment of COVID-19. Nutrients. 2021 Apr 1;13(4):1172. doi: 10.3390/nu13041172. PMID: 33916257; PMCID: PMC8065688. This article is not listed in the bibliography.

Please rephrase line 298-303. Also line 343-350.

Author Response

Point 1: Data from the literature should be better consulted. For example, what does the current meta-analysis bring new compared to that of 2022: Gavrielatou E, Xourgia E, Xixi NA, Mantelou AG, Ischaki E, Kanavou A, Zervakis D, Routsi C, Kotanidou A, Siempos II. Effect of Vitamin C on Clinical Outcomes of Critically Ill ients With COVID-19: An Observational Study and Subsequent Meta-Analysis. Front Med (Lausanne). 2022 Feb 11;9:814587. doi: 10.3389/fmed.2022.814587. PMID: 35223911; PMCID: PMC8873176.

Response 1: Thanks for the reviewer’s thoughtful comment. As the reviewer point out, we checked the referred article and added two articles; Gavrielatou et al.(2021) and Zheng et al.(2021). Also, we updated the latest review to June 2022 using Embase database instead of Google for a more detailed search and revised them on lines 118 and Figure 1.

Point 2: Also please see information from: Milani GP, Macchi M, Guz-Mark A. Vitamin C in the Treatment of COVID-19. Nutrients. 2021 Apr 1;13(4):1172. doi: 10.3390/nu13041172. PMID: 33916257; PMCID: PMC8065688. This article is not listed in the bibliography.

Response 2: In the article informed by the reviewer, we already included meta-analysis except for three articles; Capone S. et al.(2020), Alamdary DH et al.(2020), and Liu XH et al.(2020). However, we deleted them according to our exclusion criteria; no comparison, irrelevant or no outcomes. Among them, Liu et al. reported mainly clinical outcomes.

Point 3: Please rephrase line 298-303. Also line 343-350.

Response 3: As mentioned in point 1, we added total six articles to the meta-analysis through research and review. Also, we made overall modifications other than what you suggested in point 3.

Round 2

Reviewer 2 Report

This manuscript has been well revised.

I support acceptance as is.

Author Response

Thank you again for your time and consideration.